



# 1 A Holocene temperature (brGDGT) record from Garba

# 2 Guracha, a high-altitude lake in Ethiopia.

Lucas Bittner[1], Cindy De Jonge[2], Graciela Gil-Romera[3,4], Henry F. Lamb[5,6], James M. Russell[7],
Michael Zech[1]
[1] Heisenberg Chair of Physical Geography with focus on paleoenvironmental research, Institute of Geography,
Technische Universität Dresden, Dresden, Germany
[2] Geological Institute, Department of Earth Sciences, ETH Swiss Federal Institute of Technology, 8092 Zurich,
Switzerland
[3] Plant Ecology and Geobotany dept., Philipps-Marburg University, Marburg, Germany.
[4] Department of Geo-environmental Processes and Global Change, Pyrenean Institute of Ecology, CSIC, Zaragoza,
Spain
[5] Department of Geography and Earth Sciences, Aberystwyth University, Aberystwyth, UK.
[6] Department of Botany, School of Natural Sciences, Trinity College Dublin, Dublin 2, Ireland
[7] Department of Geological Sciences, Brown University, USA
*Correspondence to*: Lucas Bittner (lucas.bittner@tu-dresden.de)





**Abstract.** Eastern Africa has experienced strong climatic changes since the last deglaciation (15,000 years ago). The
driving mechanisms and teleconnections of these spatially complex climate variations are yet not fully understood.
Although previous studies on lake systems have largely enhanced our knowledge of Holocene precipitation variation
in eastern Africa, few such studies have reconstructed the terrestrial temperature history of eastern Africa from lake
archives. Here, we present (i) a new branched glycerol dialkyl glycerol tetraether (brGDGT) temperature calibration
that includes Bale Mountain surface sediments and (ii) a quantitative record of mean annual temperature (MAT) over
the past 12 cal ka BP using brGDGTs in a sediment core collected from Garba Guracha (3950 m a.s.l.) in the Bale
Mountains. After adding Bale Mountain surface sediment (n=11) data to the existing East African lake dataset,
additional variation in 6-methyl brGDGTs was observed, which necessitated modifying the MBT'$_{5ME}$ calibration by
adding 6-methyl brGDGT IIIa' (resulting in the MBT-Bale Mountain index, $r^2$=0.93, p<0.05). Comparing the MBT'$_{5ME}$
and the new MBT-Bale Mountain index, our high altitude Garba Guracha temperature record shows that significant
warming occurred shortly after the Holocene onset. The temperature increased by more than 3.0 °C in less than 600
years. The highest temperatures prevailed between 9 and 6 cal ka BP, followed by a temperature decrease until 1.4 cal
ka BP. The reconstructed temperature history is strongly linked to supraregional climatic changes associated with
insolation forcing and the African Humid Period (AHP), as well as with local anomalies associated with catchment
deglaciation and hydrology.

Keywords: paleolimnology; MAT; brGDGT, calibration, palaeoclimatology, eastern Africa
**1. INTRODUCTION**
The severity of the current climate change and its global implications have been widely discussed following the latest
report from the Intergovernmental Panel for Climate Change (IPCC) (IPCC, 2021). This highlights the need for the
scientific community to use palaeoclimate to estimate climate baseline conditions prior to human impact on climate
(Neukom et al., 2019). Although palaeoclimatology has become a central discipline in understanding current climate
variability (Thompson et al., 2002), important areas of the planet remain understudied. A partial understanding of
global climate complexity can lead to biased views of natural systems (Hughes et al., 2021). This is the case for the
African continent in general and northeastern Africa in particular. Current climatic conditions in eastern Africa vary
significantly due to its complex topography and the influence of the Intertropical Convergence Zone (ITCZ), the Indian



Monsoon and the El Niño-Southern Oscillation (ENSO). All of these affect temperature and the distribution, amount
and timing of rainfall in the region, resulting in a wide range of climatic conditions from the warm, dry and semi-arid
conditions of northern Kenya, south-eastern Ethiopia, Djibouti and Somalia to the cool, humid conditions of the
western highlands (Hove et al., 2011; Nicholson, 2017; Lyon and Vigaud, 2017).

There is clear evidence indicating that, since the last glacial period, northern and eastern Africa experienced severe
climatic changes (Tierney et al., 2008, 2011a, 2017, 2013; Loomis et al., 2015; Wagner et al., 2018). Three major
climate events are the post-glacial warming (~15 ka BP), the hydrological variability during the African Humid Period
(AHP) (15 -5 ka BP) (deMenocal et al., 2000), that lead to the greening of the Saharan Desert (Blom et al., 2009), and
the drying period at the beginning of the Meghalayan (4.2 ka BP) (Bini et al., 2019). The intensity and the timing of
these climatic changes varied regionally over northern and eastern Africa (Castañeda et al., 2016). While the driving
mechanisms and the regional differences are complex and not fully understood, evidence supports the view that
climatic changes in northern and eastern Africa were connected across the northern hemisphere (Tierney et al., 2013;
Tierney and Russell, 2007; Otto-Bliesner et al., 2014). These complex teleconnections and their global impact support
the importance of understanding long-term climate drivers in eastern Africa. Such knowledge will lead to better
assessments of the impacts and potential mitigation of the current and future climate change scenarios in this world's
understudied yet critical region.
While several studies have reconstructed the precipitation history in Northern and eastern Africa over the last 15 cal
ka BP (Bittner et al., 2021; Costa et al., 2014; Jaeschke et al., 2020; Junginger et al., 2014; Morrissey and Scholz,
2014; Tierney et al., 2011a; Trauth et al., 2018; Wagner et al., 2018), only a few have reconstructed the regional
temperature history in northeastern Africa (Castañeda et al., 2016; Morrissey et al., 2018; Berke et al., 2012a; Loomis
et al., 2017, 2012, 2015; Tierney et al., 2008, 2016). Moreover, there is a lack of terrestrial temperature reconstructions,
especially in the high altitudes and the Horn of Africa.
For terrestrial archives, different methods have been developed and applied based on pollen, chironomids, and lipid
biomarkers (Cheddadi et al., 1998; Wu et al., 2007; Chevalier and Chase, 2015; Bonnefille et al., 1992; Eggermont et
al., 2010; Schouten et al., 2007). Over the last 15 years, an innovative approach for temperature reconstructions
emerged based on branched glycerol dialkyl glycerol tetraethers (brGDGTs), membrane-spanning bacterial lipids
(Damsté et al., 2000). Several calibration studies in different settings (i.e. soils and lakes) have shown a correlation



between brGDGT abundances and mean annual air temperature (MAT) (e.g. De Jonge et al., 2014; Dearing Crampton-
Flood et al., 2020; Russell et al., 2018; Weijers et al., 2007). These calibrations have been successfully used to
quantitatively reconstruct continental temperature in marine river outflow and lacustrine sediments and terrestrial
archives such as loess sequences and paleosoils (Loomis et al., 2015, 2017; Morrissey et al., 2018; Schreuder et al.,
2016; Zeng and Yang, 2019; Garelick et al., 2022). Recently, global calibrations have been developed that suit cooler
high-latitude lakes better (Martínez-Sosa et al., 2021; Raberg et al., 2021).
The phylogenetic breadth of the brGDGT-producing bacteria is still poorly constrained, although members from the
phylum Acidobacteria have been proposed to produce brGDGTs both in cultures and in the environment (Sinninghe
Damsté et al., 2018; De Jonge et al., 2019, 2021). Originally, Weijers et al. (2007b) found that the methylation (MBT)
and cyclization of branched tetraethers (CBT) correlate with the measured mean annual air temperature (MAT) and
pH values. Following the analytical separation of 5 and 6 methyl isomers, De Jonge et al. (2014) developed a new
modified MBT'$_{5ME}$ ratio. This resulted in a revised calibration that removed the pH dependence affecting the
MBT/MAT correlation and improved the accuracy of MAT reconstructions in terrestrial/soil archives. As brGDGT
distributions recovered from lake sediments showed a different MAT dependence compared to soils, Russell et al.
(2018) developed a MBT'$_{5ME}$ temperature calibration for lake sediments in eastern Africa. However, compared to the
dataset of Russell et al. (2018), the brGDGT distribution of some Bale Mountain lake surface sediments are unique
(Baxter et al., 2019). Although the MBT'$_{5ME}$ calibration by Russell et al. (2018) is a valuable supra-regional metric for
reconstructing lake temperature, an adjusted calibration might better account for local conditions in the Bale region.
In this study, we aim to (i) compare brGDGT distributions from lake surface sediments of the Bale Mountains (n=11)
(Baxter et al., 2019) with the eastern African dataset (Russell et al., 2018), (ii) develop a new ratio that captures the
unique variation in the Bale Mountains and compare the accuracy of this calibrated ratio with the MBT'$_{5ME}$, and (iii)
reconstruct the first Horn of Africa high altitude paleotemperature record in the Bale Mountains using the sedimentary
record of Garba Guracha (3950 m a.s.l.) and (iv) compare this Garba Guracha temperature record with other records
in the region.



## 2. REGIONAL SETTINGS

### 2.1 Study area

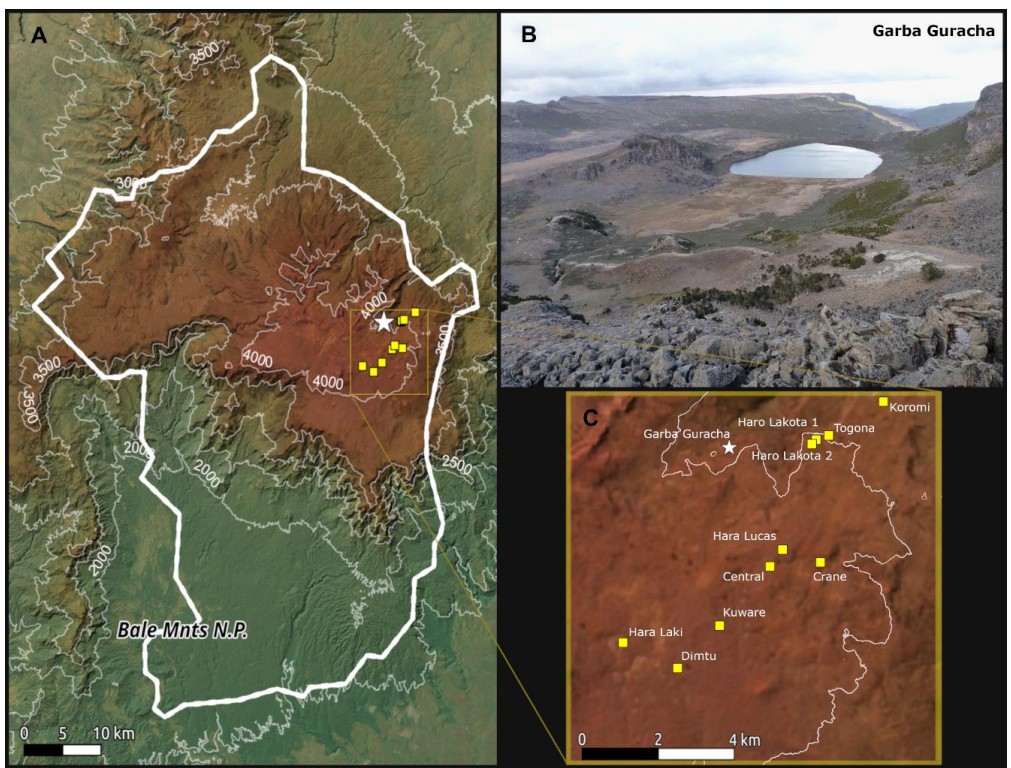

Figure 1: Location of the study area. (A) Bale Mountain National Park (thick white line), (B) a northeastward view over the glacial cirque of the Garba Guracha catchment (Bittner et al., 2021), and (C) Bale Mountain lakes in the dataset (yellow) - The map was created by the authors using QGis 3.24 Tisler. All map layers are CC-by-SA v4.0, Image is from Bing Image / DigitalGlobe © Microsoft, DEM is from NASA/JPL SRTM (http://www.jpl.nasa.gov/srtm/) and the Bale Mountains National Park boundaries are from © OpenStreetMap contributors 2019. Distributed under the Open Data Commons Open Database License (ODbL) v1.0.

Garba Guracha (6.875781N, 39.878075E; Fig. 1) and all other lakes in this study are located east of the Main Ethiopian Rift in the Bale Mountains of the Bale-Arsi Massif. More specifically, they are situated on the Sanetti Plateau, the highest plateau in the Bale Mountains, between ~3800 to ~4200 m.a.s.l. and an area of 600 km$^2$ (Osmaston et al., 2005). Solidified horizontal lava consisting of tuffs with rhyolites, alkali basalt, and trachyte formed the volcanic plateau (Uhlig and Uhlig, 1991; Williams, 2016). The plateau and the valleys were partially glaciated at the Last Glacial Maximum (Groos et al., 2021, 2020; Osmaston et al., 2005; Ossendorf et al., 2019). The glacial cirque Garba





Guracha was first mentioned by Werdecker (1962) and was also described in depth by Umer et al. (2007) and Tiercelin
et al. (2008). With a maximum water depth of 6 m and a very small catchment area, the lake is located at 3950 m above
sea level. (0.15 km$^2$; Fig. 1). The bedrock of the catchment is carbonate-poor (Löffler, 1978; Uhlig and Uhlig, 1991).
An outlet towards the Togona Valley is present during the rainy season at the lake's northern end. A swampy alluvial
plain fed by multiple springs stretches along the lake's southern shore.

**2.2 Climate**
The climate of the Bale Mountains varies spatially and temporally, affected by the orographic differences in altitude,
a north-south exposure and by changing atmospheric air mass movements over the course of the year (Kidane et al.,
2012; Uhlig and Uhlig, 1991). The Bale Mountains experience a four-month dry season (November to February) and
a long wet season with complex orographic rainfall patterns (March to October) (Woldu et al., 1989; Kidane et al.,
2012). The complexity of the rainfall pattern is associated with the convergence of northeast and southwest winds due
to the northern and southern location of the ITCZ between June and September and between October and March,
respectively (Tiercelin et al., 2008; Kidane et al., 2012). The Equatorial Westerlies and the Indian Ocean monsoon act
as two moisture sources for the precipitation in the Bale Mountains (Miehe and Miehe, 1994; Uhlig, 1988). With 1000-
1500 mm per year, the southern part of Bale Mountain experiences the highest precipitation amount, whereas the
northern region, including Garba Guracha, only receives 800-1000 mm (Woldu et al., 1989). Temperatures vary
seasonally, with the lowest temperatures in the dry season and the highest temperatures in the rainy season (Hillman,
1988). The Afro-Alpine regions, including the Sanetti Plateau, are characterized by diurnal temperature differences
between day and night (-15 to +26°C) (Hillman, 1988). Across the Bale Mountains, climate data has been collected
since 2017 with a mean annual temperature of 4.9 °C (max. 6°C; min. 3.4 °C) at the Angesso Station, located at the
same altitude 4 km northeast of Garba Guracha. The mean annual temperature at Garba Guracha is 5.4°C (Baxter et
al., 2019).





## 3. MATERIAL AND METHODS

### 3.1. Material and Sampling

In this study, we used the data of 76 surface sediment samples from eastern African lakes. These lake sediments, located mainly in Ethiopia, Uganda and Kenya, were analyzed, and the results were published by Loomis et al. (2014, 2011, 2012), Russell et al. (2018), Eggermont et al. (2011) and Baxter et al. (2019). The environmental data for the 11 lakes in the Bale Mountains were published by Eggermont et al. (2011) and Baxter et al. (2019), and the corresponding MAT is based on a calculated lapse rate supported by local climate station data (Loomis et al., 2012; Russell et al., 2018).

At the Garba Guracha site, two overlapping sediment cores were retrieved in February 2017, at a water depth of 4.8 m using a Livingstone piston corer. A maximum sediment depth of 1550 cm was reached, covering an organic matter-rich upper section (0-900 cm) and an organic matter-poor bottom one (900-1550 cm). This study focuses on the last 12.3 cal ka BP covering the 0-950 cm, with a mean sedimentation rate of 15 years/cm (more details on sediment properties and chronology can be found in Bittner et al. (2020)). We sampled at contiguous 10 cm intervals (average ~100 years of sedimentation). Thirty-five samples were selected for brGDGT analyses.

### 3.2 Sample preparation and analysis

The total lipid extracts (TLE) of the surface sediment samples were extracted using an accelerated solvent extractor (ASE) with dichloromethane:methanol in a ratio of 9:1 (Loomis et al., 2012). The brGDGTs were purified and separated according to their polarity. The samples were quantified following the method described by Huguet et al. (2006).

The TLE of the downcore sediments was obtained using a soxhlet system by constant rinsing (24h) with solvent (dichloromethane:methanol in a ratio of 9:1). After rotary evaporation, the TLE was redissolved in $n$-hexane and transferred onto a pipette column filled with aminopropyl silica gel (Supelco, 45 um). Solvents of increasing polarity ($n$-hexane, dichloromethane/methanol 2:1; diethyl ether/acetic acid 19:1) were used to selectively elute the fractions of the TLE (nonpolar fraction A; two polar fractions B and C, including brGDGTs). Fraction B contained 98-99%, while fraction C contained 1-2% of all brGDGTs. All results refer to the brGDGTs contained in fraction B. Before measurement, a $C_{46}$ brGDGT standard was added, and the extract dried, redissolved in n-hexane/isopropanol (99:1) and filtered using a 0.45 um polytetrafluoroethylene (PTFE) filter. The measurements of the GDGTs (dissolved in n-



hexane/IPA (99:1)) were done at ETH Zurich using a high-performance liquid chromatograph (Agilent 1260) coupled
to a quadrupole mass spectrometer configured for atmospheric pressure chemical ionization (HPLC-APCI-MS). The
separation of the GDGTs was achieved by two silica columns at 45°C (modified after Hopmans et al. (2016)) with a
flow rate of 0.2ml/min and an injection volume of 10 µl. Compound-peak integrations of $m/z$ 1292, 1050, 1048, 1046,
1046, 1034, 1032, 1022, 1020, 1018 and 744 were performed according to previously published methods (Hopmans
et al., 2016).
**3.3 BrGDGTs – structure, statistical methods and proxy calculation**
BrGDGTs can be present as tetra- (I), penta- (II), or hexamethylated compounds with different numbers of cyclopentyl
moieties (none (a), one (b), or two (c)). The outer methyl group can be positioned on the α and/or C5 (5-methyl
compounds) or C6 (6-methyl compounds, indicated by a prime notation) location (De Jonge et al., 2014). To interpret
the GDGT composition of the samples, we used the BIT, MBT', MBT'$_{5ME}$, and CBT' (Table 1).
We calculated the BIT index following the equation of Hopmans et al. (2004):
$BIT index = (Ia + IIa + IIIa + IIa' + IIIa')/(Ia + IIa + IIIa + IIa' + IIIa' + crenarchaeol)$      [Eq. 1]
De Jonge et al. (2014) showed that the MBT' ratio (Peterse et al., 2012) contains 5- and 6-methyl compounds that are
explicitly mentioned here:
$MBT' = (Ia + Ib + Ic)/(Ia + Ib + Ic + IIa + IIa' + IIb + IIb' + IIc + IIc' + IIIa + IIIa')$      [Eq. 2]
By removing the 6 methyl isomers from the equation, De Jonge et al. (2014) improved the temperature calibration
further:
$MBT'_{5ME} = (Ia + Ib + Ic)/(Ia + Ib + Ic + IIa + IIb + IIc + IIIa)$      [Eq. 3]
The cyclization of branched tetraethers (CBT') is calculated following the equation from De Jonge et al. (2014a):
$CBT' = {}^{10}log (Ic + IIa' + IIb' + IIc' + IIIa' + IIIb' + IIIc')/(Ia + IIa + IIIa)$      [Eq. 4]
The fractional abundance of any individual brGDGT compound (i) was defined as:
$f(i) = i/(Ia + Ib + Ic + IIa + IIa' + IIb + IIb' + IIc + IIc' + IIIa + IIIa' + IIIb + IIIb' + IIIc + IIIc')$[Eq. 5]



### 3.4 Quantitative data analyses

Numerical analyses in this paper have been performed with Excel and R 4.1.0 (R Core Team, 2021). Results are displayed using the arithmetic mean and standard deviations using the notation ±. To explore the correlation level between brGDGTs and MAT, we used linear regressions and the reported Pearson correlation values (r2), where significant correlations were considered when the p-value < 0.05. We performed a Principal Component Analysis (PCA) of brGDGTs from i) the calibration dataset and ii) the Garba Guracha record, based on standardized and scaled fractional abundance. The ordination methods provide a simple yet effective way to visualize the variability within the distribution of the brGDGTs. PCA was performed with the R package *factoextra* (Kassambara and Mundt, 2020).





## 4. RESULTS

### 4.1 BrGDGT patterns of surface sediments from lakes in the Bale Mountains

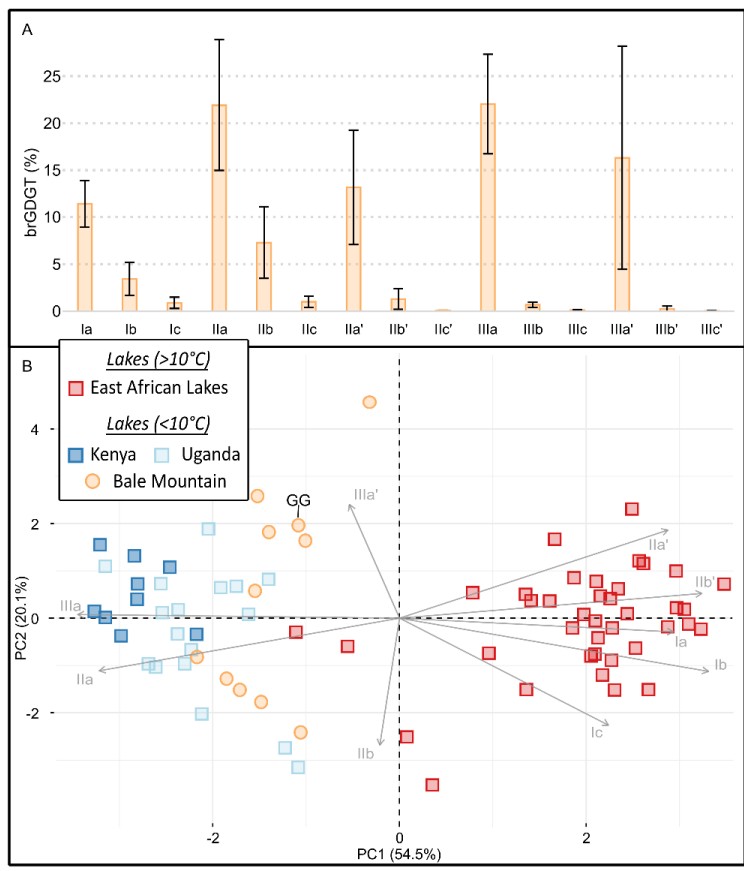

Figure 2: (A) Barplot of average brGDGT percentages in Bale Mountain lake surface sediments (Baxter et al., 2019), with standard deviation plotted as error flags and (B) PCA of brGDGTs of eastern African lakes with regional pattern; data from Russell et al. (2018) and Baxter et al. (2019) - lakes >10°C (purple) and lakes <10°C (red, turquoise and green); Garba Guracha (GG).

To frame the downcore variation in Garba Guracha in the current environmental settings, we have expanded the dataset of Russel et al. (2018) by 11 Bale Mountain lake surface sediment samples (Table S1 and S2) (data from Baxter et al., 2019). Due to the missing values of IIc, IIc', IIIb, IIIb', IIIc and IIIc' in the dataset of Russell et al. (2018), we excluded these isomers in the Bale Mountain data from the PCA to compare the datasets (see the supplementary Figure S1 for a PCA including all isomers). The highest fractional abundances of brGDGTs in these surface sediments are (i) IIIa with a mean of 22% (±5), (ii) IIa with a mean of 22% (±7) and (iii) IIIa' with a mean of 16% (±12) (Fig. 2A, Table S2).





The PCA of brGDGTs shows some differences between the East African lake dataset and the Bale Mountain lakes
(Fig. 2B). IIIa and IIa have negative loadings, and IIa', IIb', Ia, Ib and Ic have positive loading on PC1. PC2 has negative
loadings from IIb and positive loading from IIIa' and IIa'. The Bale Mountain lakes have a negative score on PC1,
consistent with their location in a cool climate. At the same time, the Bale Mountain lakes have a wider dispersion on
PC2 than the East African lake dataset. On PC2, a decrease of 5ME cyclopentane of brGDGTs IIb and an increase of
6 ME brGDGT IIIa' is visible in some of the Bale Mountain lakes, including Garba Guracha.
Compared to similar high altitude lakes (above 3500 m) in eastern Africa (East African lake dataset (Russell et al.,
2018)), the percentage of IIIa and IIa is lower, and the IIIa' and IIa' is higher in the Bale Mountain lakes (Fig 3).
Interestingly, the combined percentage of these 5 and 6 methyl isomers is similar (Fig. 3 A).

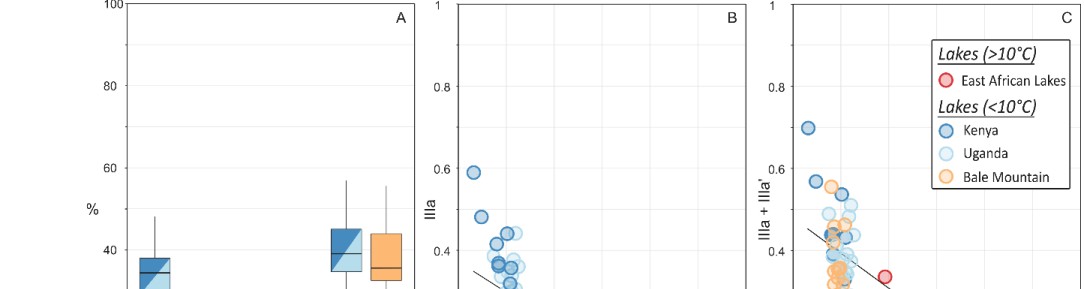


Figure 3: (A) Abundance of IIIa and IIIa' in percent; (B) Linear correlation between IIIa and MAT ($r^2 = 0.78$) and (C) IIIa + IIIa' to MAT ($r^2 = 0.82$)
- data from Russell et al. (2018) - lakes >10°C (purple) and lakes <10°C (red and green); Bale Mountain lakes (Baxter et al., 2019) (turquoise).

We hypothesize that the 6-methyl compound (IIa' and IIIa') is produced instead of their 5-methyl counterparts (IIa and
IIIa) in some of the Bale Mountain lakes. Moreover, in the East African lake dataset, the correlation of IIIa to MAT
($r^2 = 0.78$) is slightly improved by adding IIIa' to $r^2 = 0.82$ (Fig 3 B and C). Although the production of IIIa' at the
expense of IIIa is poorly understood in lacustrine settings, it clearly has the potential to influence MBT'$_{5ME}$ values.
Indeed, MBT'$_{5ME}$ values of the Bale Mountain lakes range from 0.20 to 0.37, with a mean of 0.24 (±0.05). As the MAT
range of Bale Mountain lakes is limited (4-5.4 °C), the range of MBT'$_{5ME}$ is larger than expected of the measured MAT





relative to similar eastern African lakes in the East African lake dataset (MBT'$_{5ME}$ = 0.17 to 0.25 with a mean of 0.22
± 0.02; MAT = 4-5.4 °C) (Russell et al., 2018). Moreover, regional variations in brGDGT isomers, especially in Bale
Mountain surface sediments, appear in the degree of cyclization (DC') and CBT' in the eastern African lake surface
sediment data (Russell et al., 2018; Baxter et al., 2019) (Fig. S2).
**4.2 BrGDGT patterns of the Garba Guracha sediment core**
In the Garba Guracha sediments, both branched and isoprenoid GDGTs are present. The BIT index ranges between
0.8 and 1 (mean=0.98, ± 0.04). Only the oldest samples (12-10 cal ka BP) have a lower BIT index value of 0.8 to 0.9
(Table S3). Tetramethylated brGDGTs in the sediment core represent on average 19.5%, pentamethylated brGDGTs
44%, and hexamethylated brGDGTs 36.5% (Table S3). The highest fractional abundances are (i) IIIa with a mean of
21% (±5), (ii) IIa with a mean of 20% (±3) and (iii) Ia with a mean of 15% (±3). The MBT'$_{5ME}$ ranges from 0.20 to
0.35 with a mean of 0.28 (±0.04) (Table S4). The CBT' ratio ranges from 0.06 to -0.54 with a mean of 0.27 (±0.18)
(Table S3).
A PCA of all downcore brGDGTs distributions (Fig. 4A) shows that the first two components explain 63.2% of the
variance. On PC1 (42.3%), all 6 methyl isomers have negative loadings, while 5 methyl isomers show positive
loadings. PC2 (20.9%) shows positive loadings of all hexamethylated brGDGTs and negative loadings of all penta-
and tetramethylated brGDGTs. The PCA reveals changes in brGDGT composition with core depth when the data
points are grouped using the following age cut-offs: (0-4.3 cal ka BP; 4.3-10.5 cal ka BP; 10.5-12.5 cal ka BP) (Fig.
4A). In phase 1 (12.5 – 10.5 cal ka BP), IIIa, IIIa' and IIa have the highest mean abundances of 30%, 17%, and 17%,
respectively. In phase 2 (10.5 – 4.3 cal ka BP), the mean abundance of IIIa and IIIa' are decreased by around 9%, while
IIa, IIb and Ia increase. In phase 3 (4.3 – 0 cal ka BP), the mean abundances of IIIa decrease by 6% further. Conversely,
the mean abundance of IIIa' increases again by 12%. The same holds true for IIa (-5%) and IIa' (+6%). The mean
abundance of Ia increases further by 3% (Fig 4B).





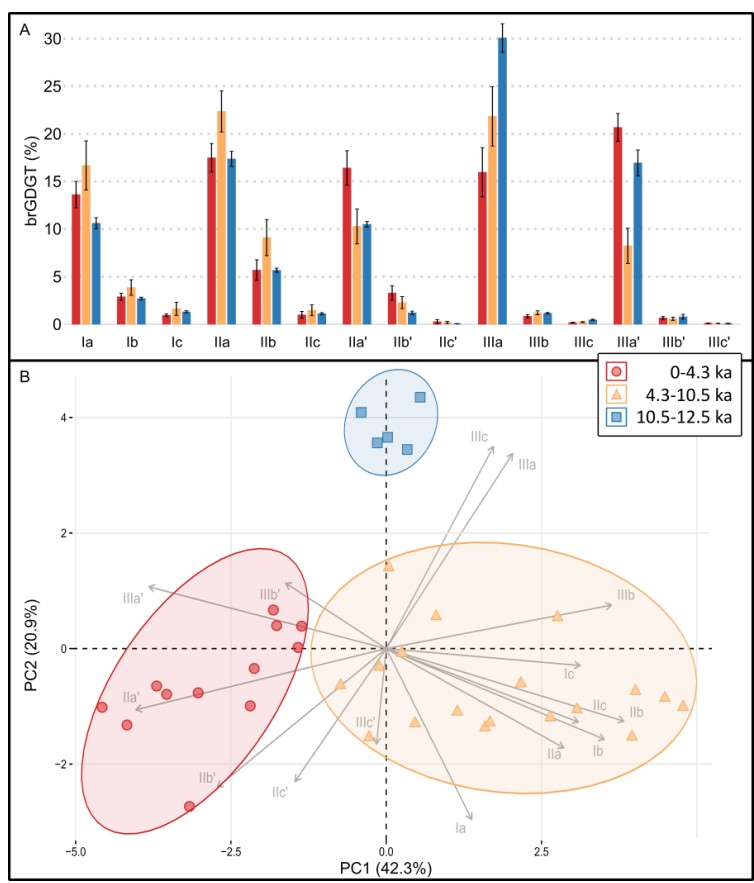

Figure 4: (A) PCA of brGDGTs of the Garba Guracha sediment core; data from 0 to 4.3 cal ka BP (red), data from 4.3 to 10.5 cal ka BP (orange) and data from 10.5 to 12.5 cal ka BP (blue); (B) barplot of average brGDGT percentages in the Garba Guracha sediment core, with standard deviation plotted as error flags.

The unusually high abundance of brGDGTs IIIa' compared to IIIa observed in surface sediments of Bale Mountains lakes (Fig. 2A) is also visible in the Garba Guracha record. The relative abundance of IIIa' varies with depth. High amounts of IIIa' appear until 10.8 cal ka BP followed by low percentage (<10%) until 4.5 cal ka BP. The highest abundance of IIIa' with up to 22% occurs after 4.5 cal ka BP until the recent past. The changing abundances of IIIa' in our record coincide with changes in CBT' (Fig. 5). The variability in the 6-methyl brGDGTs reflects the largest part of the variation in this dataset, reflected by the good agreement ($r^2$= 0.77, p<0.001) between the fractional abundance of brGDGTs IIIa' and the sample loadings on PC1.




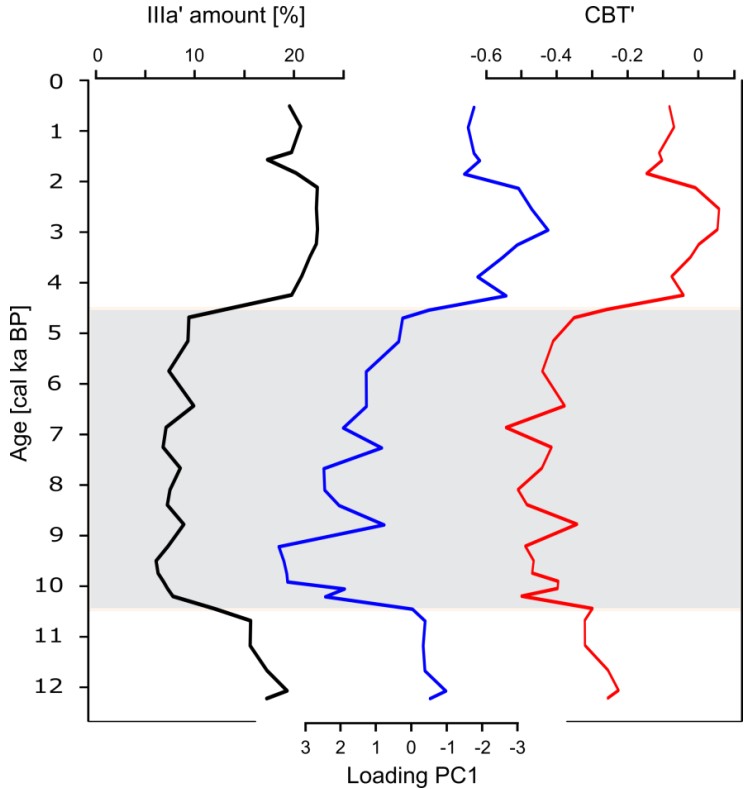

270

Figure 5: Downcore functions for IIIa' amount, the PC1 loading and CBT' of the Garba Guracha brGDGT record.

272



## 5. DISCUSSION

### 5.1 Possible MAT calibration functions inferred from the expanded east African surface sediment dataset

*Table 1: Temperature calibrations – Ratios, calibration dataset, $r^2$, and root-mean-square-error (RMSE) in °C*

| Ratio | | Calibration dataset | $r^2$ | RMSE °C |
|---|---|---|---|---|
| MBT'$_{5ME}$<br>(Ia+Ib+Ic)/(Ia+Ib+Ic+IIa+IIb+IIc+IIIa) | | EAL (n=65) | 0.92 | 2.41 |
| | | EAL$_{BM}$ (n=76) | 0.92 | 2.41 |
| MBT'$_{5ME}$ + IIIa'<br>(Ia+Ib+Ic)/(Ia+Ib+Ic+IIa+IIb+IIc+IIIa+IIIa') | [Eq. 6] | EAL$_{BM}$ (n=76) | 0.93 | 2.38 |
| MBT'$_{5ME}$ + IIa'&IIIa'<br>Ia/(Ia+IIa+IIIa+IIa'+IIIa') | [Eq. 7] | EAL$_{BM}$ (n=76) | 0.84 | 3.48 |
| Simplified MBT'$_{5ME}$ + IIIa'<br>Ia/(Ia+IIa+IIIa+IIIa') | [Eq. 8] | EAL$_{BM}$ (n=76) | 0.91 | 2.59 |

We added the GDGT distribution data of 11 surface sediments from Bale Mountains lakes (Baxter et al., 2019) to the existing data of Russell et al. (2018) and applied the MBT'$_{5ME}$ calibration (Table 1). Here, the original dataset (n = 65) is referred to as East African Lakes "EAL", while the extended dataset (n = 76) is referred to as East African Lakes + Bale Mountain lakes (EAL$_{BM}$). The linear correlation between the MBT'$_{5ME}$ and MAT was almost identical after adding the 11 Bale Mountain lake samples (EAL $r^2$= 0.92, EAL$_{BM}$ $r^2$ = 0.93). To test whether the unique brGDGT distribution in some Bale Mountain lakes (Fig. 2) affected the temperature correlation, we applied various calibrations to account for the increased abundance of IIIa' (and to a lesser extent IIa'). Including these compounds is *a priori* supported by global scale calibrations, as Raberg et al. (2021) showed that a global temperature calibration with a modified MBT' ratio that includes IIIa' and IIa' (Table 6, Eq. 7) correlates with month above freezing (MAF) in lakes ($r^2$ = 0.90, RMSE= 2.18) almost as well as the unmodified MBT'$_{5ME}$ ratio. In the EAL$_{BM}$ dataset, the application of this ratio has a lower $r^2$ of 0.84 and a higher RMSE of 3.48°C compared to the MBT'$_{5ME}$ (Table 1: Eq. 7). As brGDGT IIIa' specifically was shown to increase in Bale Mountain sediments and improved the correlation with MAT (Fig. 3B and C), we investigated alternative ratios that incorporate this compound but exclude IIa'. Table 1 and Fig. 6 summarize the correlation coefficients of the MBT'$_{5ME}$ ($r^2$ = 0.92, RMSE of 2.41°C), an MBT'$_{5ME}$ ratio that includes IIIa' (Eq. 6) with $r^2$ = 0.93 and RMSE of 2.38°C and the simplified ratio that includes only the major brGDGTs compounds (Eq. 8: $r^2$ = 0.91 and an RMSE of 2.59°C).
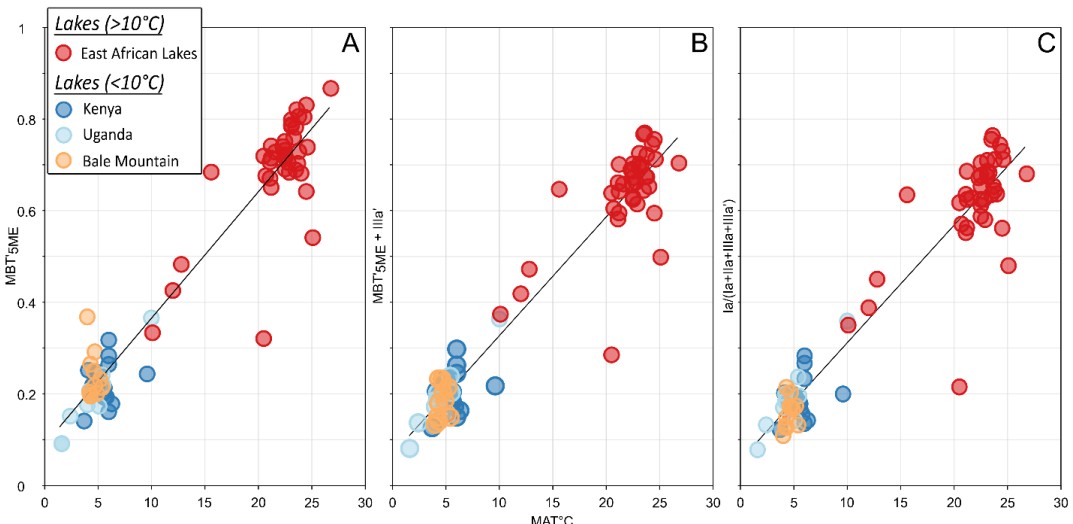


Figure 6: Correlations EAL$_{BM}$ datasets, (A) MBT'$_{5ME}$ ($r^2$ = 0.92; RMSE of 2.41); (B) MBT'$_{5ME}$ + IIIa' ($r^2$ = 0.93; RMSE of 2.38).); (C)

Ia/(Ia+IIa+IIIa+IIIa') ($r^2$ = 0.91; RMSE of 2.59) - data from Russell et al. (2018) - lakes >10°C (purple) and lakes <10°C (red and green); Bale

Mountain lakes (Baxter et al., 2019) (turquoise).

The results of Ia/(Ia+IIa+IIIa+IIIa') (MAT= -0.773 + 35.646 x Ia/(Ia+IIa+IIIa+IIIa')) and MBT'$_{5ME}$ + IIIa' (MAT= -
1.4734 + 35.777 x MBT'$_{5ME}$ + IIIa') calibrations are very similar ($r^2$=0.97) (Fig. 7, purple and green curves,
respectively).  Therefore, we discuss only the best performing MBT'$_{5ME}$ (MAT= -1.8299 + 33.304 x MBT'$_{5ME}$) and the
MBT'$_{5ME}$ + IIIa' calibration developed using the EAL$_{BM}$ dataset to the downcore distributions.

**5.2 Paleotemperature reconstructions for the Garba Guracha sedimentary record - comparison of the different**
**calibrations**

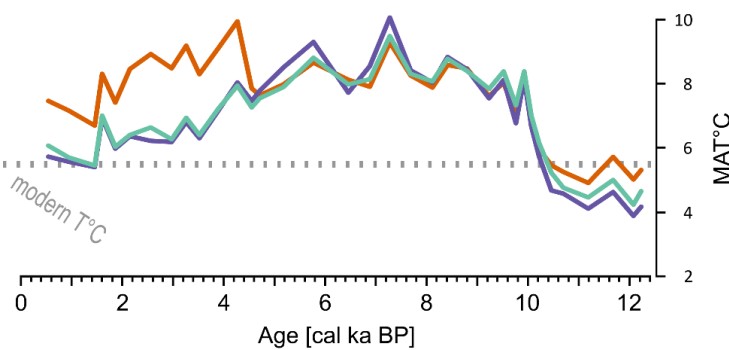


Figure 7: Reconstructed temperatures of the Garba Guracha sedimentary record. MBT'$_{5ME}$ (orange); Ia/(Ia+IIa+IIIa+IIIa') (purple); MBT'$_{5ME}$ + IIIa'

(turquoise).

We evaluate the downcore trend in GG sediments to compare the performance of both calibrations, revealing periods
of agreement (10-4.2 ka BP) and a period of temperature offset (since 4.2 ka BP). Using established and newly
developed ratios and calibrations (Russell et al., 2018; this paper) resulted in similar absolute values and comparable
temperature trends, principally in the early and mid-Holocene. Reconstructed temperatures range from 4.9 to 10.0°C
(MBT'$_{5ME}$) and 4.2 to 9.5°C (MBT'$_{5ME}$ + IIIa') (Fig. 7). Despite a slightly different range in temperature (4.4 and 5.3
°C), the trends of both calibrations are similar between 12 and 4.7 cal ka BP. The lowest MATs (< 5°C) occurred
between 12.2 cal ka BP (950 cm) and 10.5 cal ka BP (800 cm). MAT increased rapidly by 3.5°C between 10.5 cal ka
BP (800 cm) and ca. 10 cal ka BP (700 cm). During the early to mid-Holocene, a thermal maximum occurred between
10 and 5.7 cal ka BP (440 cm), with the highest MAT values reaching ca. 10°C.
At ~6.5 cal ka BP, the MAT decreased for both calibrations. The temperature drop coincides with organic-poor layers
in the sediment core formed during a drought, concurring with low monsoonal intensity (Bittner et al., 2020). While
drought phases are not directly linked to changing temperatures, in the monsoonal systems of eastern Africa,
temperature and precipitation seem to be more connected (Costa et al., 2014; Loomis et al., 2015).
A strong offset between the calibrations appeared at 4.2 cal ka BP. Using the MBT'$_{5ME}$, we reconstruct a sudden
temperature rise (Fig. 7) that contrasts with the temperature decrease when using the MBT'$_{5ME}$ + IIIa' calibration at a
moment when temperatures are expected to decrease in phase with insolation (Fig. 7). The offset coincides with a
known drought phase and is accompanied by shifts of many proxies (TOC, $\delta^{13}$C, TOC/N, *Erica spp*, charcoal) in the



Garba Guracha record (Bittner et al., 2020; Gil-Romera et al., 2019). The changing conditions in the Garba Guracha
catchment during this drought phase, especially the decline of the *Erica* shrubland (Gil-Romera et al., 2019), could
have resulted in an increase in the lake water pH. A change in the lake water chemistry is supported by a decrease in
CBT' ratio (Fig. 5). In lakes on a global scale, CBT' correlates with pH and conductivity (Raberg et al., 2021). In the
last years, studies have suggested that the change in brGDGT composition captured by the CBT' may change due to
shifting bacterial communities in soils and lakes (De Jonge et al., 2019; van Bree et al., 2020; Weber et al., 2018).
Previously, pH-dependent community changes have been shown to affect MBT'$_{5ME}$ values in soils (De Jonge et al.,
2021), and we propose that a similar effect can be seen in Garba Guracha.
Hence we suggest that MBT'$_{5ME}$ systematically overestimates the temperatures of Garba Guracha during the late
Holocene after 4 cal ka BP. A systematic offset is further supported by continuously and similarly decreasing
reconstructed temperatures using both calibrations until the top of the core with a shared maximum at 150 cm (1.6 cal
ka BP). We suggest that the production of IIIa' at the expense of IIIa is increased during dryer intervals, possibly caused
by a change in lake water chemistry and/or bacterial communities. We conclude that a temperature calibration
including IIIa' allows to reconstruct MAT in Garba Guracha sediments more accurately, as it accounts for the unique
and variable production of IIIa' in Bale Mountain lakes.

**5.3 Paleotemperature reconstructions for the Garba Guracha sedimentary record – regional comparison**

5.3.1. Deglacial warming
Overall, the recorded temperature trends in Garba Guracha are in phase with northern summer insolation variability
(Fig. 8). This is reasonable because air temperature and insolation are closely connected (Huybers, 2006). However,
the coldest MATs (<5°C) were recorded before 10.5 cal ka BP even though the northern hemisphere summer (20°N)
insolation maxima occurred already 12 cal ka BP (Fig. 8). Tiercelin et al. (2008) argue that in Garba Guracha, ice
remained in the catchment until ~10 cal ka BP due to topographical conditions, especially the north-facing exposition
of the valley. The remaining ice in the basin might have (i) reduced the temperature of the lake water by inflow of cold
melt water and (ii) buffered the warming caused by increasing insolation. Indeed, rising temperatures were recorded
in other eastern African records as early as 14 cal ka BP (Lake Tana) (Loomis et al., 2015; Tierney et al., 2016).
Similar to Lake Tana, but ~4000 years later, MAT (°C) in the Garba Guracha record experienced an abrupt increase
of ca. 3.5°C in just ca. 600 years, from 10.5 to 9.9 cal ka BP. Simultaneously with the rise in temperature, Bittner et



al. (2021) found an increase in P/E, indicating higher moisture availability based on depleting values of reconstructed
$\delta^{18}O_{\text{lake water}}$. At Lake Tana, Loomis et al. (2015) and Costa et al. (2014) attribute a similar connection between warmer
temperature and depleted water isotopes ($\delta^2H$) since 13.8 cal ka BP to the penetration of warm Congo Basin air masses
resulting in weaker easterly trade winds and a strengthening of the southwesterly winds and the Somali Jet. The con-
nection between Congo Basin air masses and eastern Africa is supported by the absence of cold temperatures associated
with the Younger Dryas (YD) in both the Congo Basin temperature record (Weijers et al., 2007a) and Lake Tana
(Loomis et al., 2015). However, in the Garba Guracha record, lower temperatures prevailed 4000 years longer than in
Lake Tana (Loomis et al., 2015). Although catchment glaciers could have caused these conditions in Garba Guracha,
the low temperatures are accompanied by a reduced sedimentation rate between 12.8 and 11.3 cal ka BP (Bittner et al.,
2020), pointing to climatic influences associated with YD times (Alley, 2000). Indeed, other records from the Horn of
Africa indicate dry conditions associated with the YD period, like Lake Ashenge (Marshall et al., 2009) and the marine
record of the Gulf of Aden (Tierney and deMenocal, 2013). Therefore, we suggest that, at least for some periods, the
climate drivers operating in the Garba Guracha region might have been different from other parts of eastern Africa.
The time lag between Lake Tana and Garba Guracha could be explained by a slow eastwards advance of the Congo
Air Boundary and different climatic conditions at the sites. However, with the current data, we are unable to precisely
distinguish between north hemisphere YD forcing, remaining ice in the lake catchment, or regional atmospheric cir-
culation change affecting the Garba Guracha record.
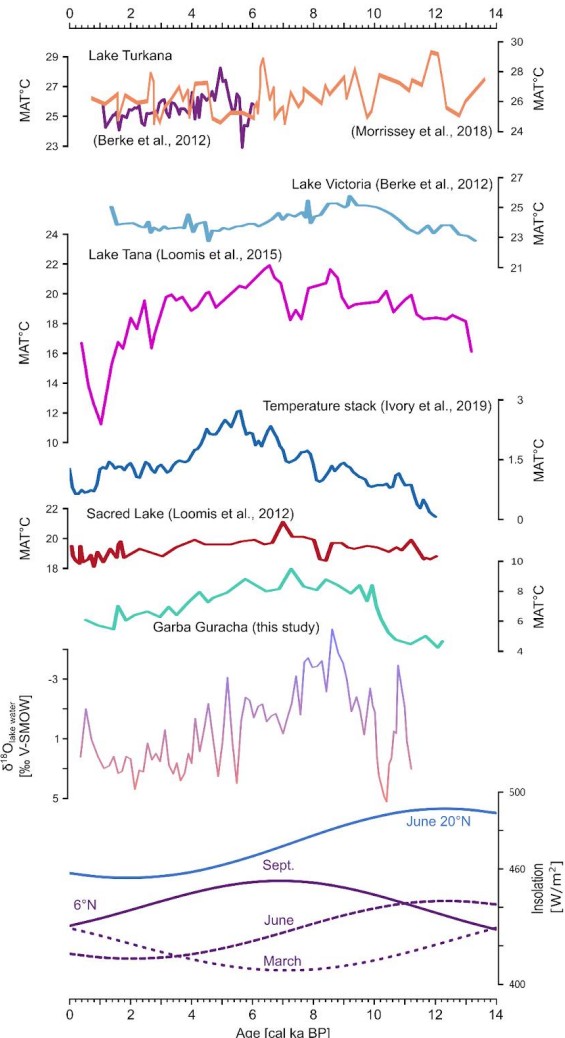


Figure 8: Comparison of records. MAT: Lake Turkana (Berke et al., 2012; Morrissey et al., 2018); Lake Victoria (Berke et al., 2012); Lake Tana (Loomis et al., 2015); eastern Africa temperature stack (Ivory et al., 2019); Sacred Lake (Loomis et al., 2012); Garba Guracha (this study); $\delta^{18}O_{lake\ water}$ as reconstructed from the aquatic sugar biomarker fucose (Bittner et al., 2021) and insolation 6°N and June 20°N (Laskar et al., 2004).

### 5.3.2. Warm temperatures during the African Humid Period in eastern Africa

Regardless of the cause, the ~10.5 cal ka BP rise in MAT is associated with an abrupt increasing moisture availability and changes of vegetation around Garba Guracha (Gil-Romera et al., 2021; Umer et al., 2007). Vegetation and fire dynamics around Garba Guracha responded dynamically to the changing climatic conditions, evidencing the sensitivity



of the afromontane-afroalpine plant communities to increasing temperature. As MAT increased between 11 and 10 cal
ka BP, the ericaceous belt expanded (Gil-Romera et al., 2021). The rising temperature and increasing P/E (Bittner et
al., 2021) were accompanied by the expansion of the afroalpine vegetation cover (Gil-Romera et al., 2021; Miehe and
Miehe, 1994). An immediate consequence of the temperature rise and increasing moisture availability was biomass
accumulation, as evidenced by the change from organic matter-poor to organic matter-rich sedimentation (Bittner et
al., 2020) and the expansion of heathlands (Gil-Romera et al., 2019). Under an increasing MAT and extending biomass,
fire activity was very intense at this time (Gil-Romera et al., 2019).
The thermal maximum of the Garba Guracha record spanned from 9 to 5.8 cal ka BP, with the highest reconstructed
temperatures occurring at 7 cal ka BP. A similar mid-Holocene thermal optimum has been recorded at Sacred Lake (7
cal ka BP) and Lake Tana (7 cal ka BP) (Fig 8). However, the highest temperatures of Lake Victoria occurred at 9 cal
ka BP, and of Lake Rutundu, Lake Malawi and Lake Tanganyika at 5 cal ka BP (Berke et al., 2012b; Loomis et al.,
2017, 2015, 2012; Powers et al., 2005; Tierney et al., 2008). At Lake Turkana, the thermal optimum occurred at 6.4
cal ka BP (Morrissey et al., 2018) or 5 cal ka BP (Berke et al., 2012a). A new temperature reconstruction from Lake
Mahoma (Garelick et al., 2022) and a temperature stack including temperature reconstructions from Sacred Lake, Lake
Malawi, Lake Tanganyika, Lake Rutundu, and the Congo Basin by Ivory and Russell (2018) showed the highest tem-
peratures between ~7 and ~4.5 cal ka BP. The timing of the highest reconstructed temperatures at these sites is not
related to greenhouse gas radiative forcing or insolation forcing (Loomis et al., 2015). Loomis et al. (2015) point out
that the Lake Tana and Sacred Lake temperature maxima lag northern hemisphere summer insolation, and Lake Ma-
lawi and Lake Tanganyika lead peak southern summer insolation. In the case of Garba Guracha, the highest tempera-
tures coincide with local maximum September insolation at the sites latitude of 6°N (Laskar et al., 2004) (Fig. 8). This
matches the suggestion of Berke et al. (2012a) that the thermal optimum of several eastern African lakes might be
determined by local solar irradiance from Sep to Dec (maximum at ~6 cal ka BP) (Fig. 8) rather than northern hemi-
sphere summer solar irradiance. The restratification processes of eastern African lakes in these months and associated
epilimnetic heating might explain the increased warming of lake water (Berke et al., 2012a). However, modelling
studies do not support this hypothesis (Dee et al., 2021).
In addition to local insolation changes, local changes in P/E could have the potential to modify the lake water temper-
ature. During the Early and Mid-Holocene, reconstructed high temperatures occurred during the African Humid Period,
accompanied by the wettest phase of Garba Guracha (Bittner et al., 2021) and rising lake levels in the region (Gasse,



2000; Junginger et al., 2014), indicating higher amounts of precipitation due to an intensification of the monsoon
system. A modelling study (Tierney et al., 2011b) proposes that during the AHP, the precipitation increase occurred
mainly in June, July, and August (JJA), shortening the duration of annual drought phases in eastern Africa. Increased
relative humidity would reduce evaporation, limiting the evaporative cooling of the lake water. Less evaporation, either
due to shorter drought phases or generally higher precipitation, would increase the temperature and cause less positive
$\delta^{18}O_{lake\ water}$ values, as suggested for Garba Guracha (Bittner et al., 2021).
The highest temperatures of the Holocene continued until 5.8 cal ka BP, interrupted only by a short drop in temperature
after 7 cal ka BP. This is in agreement with the Sacred Lake temperature record (Loomis et al., 2012). Lake Tana
experienced a shift towards colder conditions a bit earlier, from 7.5 to 7 cal ka BP (Loomis et al., 2015).
5.3.3. cooling in the Late Holocene
After 5.8 cal ka BP, the MAT continuously decreased by ~3.6°C until recent times, coinciding with the summer inso-
lation decline and decreasing temperatures of equatorial lakes (Ivory and Russell, 2018), Lake Tana (Loomis et al.,
2015) and the marine Gulf of Aden record (Tierney et al., 2016). The general decreasing temperature trend is also
supported by $\delta^{18}O_{lake\ water}$, pollen and charcoal results showing a decrease in moisture availability and fire activity
(Bittner et al., 2021; Gil-Romera et al., 2019). Furthermore, an upwards shift of the lower and dry forests during this
time reinforces the idea of more intense evapotranspiration due to the decrease in moisture availability (Gil-Romera et
al., 2021). A drop in TOC and decreasing $\delta^{13}C$ values (Bittner et al., 2020) support overall shifting catchment condi-
tions.
During the last two thousand years, we observed that the increasing temperature trend concurred with an abrupt in-
crease in the main woody communities and enhanced fire activities around Garba Guracha (Gil-Romera et al., 2021).
However, we cannot discard human influence favouring both woody encroachment and fire activity.
The strong connection of temperature, P/E and insolation across the Holocene shows that the Garba Guracha temper-
atures might have been affected by local radiation, possibly in interplay with insolation-driven atmospheric circulation
changes and their impacts on air mass source, cloud cover and evaporation. As current global warming continues, the
intense warming of landmasses could lead to a major and complex restructuring of the atmospheric circulation system
in the future, affecting eastern Africa and possibly even larger regions beyond via teleconnections.





## 6. CONCLUSIONS

The eastern African climatic history is spatially very diverse, and the driving mechanisms are complex and not fully understood. In eastern Africa, temperature reconstructions are generally sparse, especially in the high altitudes of the Horn of Africa. In this study, we used brGDGT from a high altitude sedimentary record of the Bale Mountains (lake Garba Guracha, Southwestern Ethiopia) to produce the first temperature reconstruction for the Horn of Africa.

The composition of brGDGT isomers in sediment records is affected by several influences, mainly by MAT, but in addition by lake water chemistry and bacterial community, resulting in locally unique brGDGT compositions. For instance, in some of the Bale Mountain lakes, the abundance of a specific isomer IIIa' is uncommonly high in surface sediments. However, the summed abundance of IIIa and IIIa' is similar to other comparable lake archives in eastern Africa. We suspect that in the case of the Bale Mountains, changes in the lake's water chemistry or bacterial community are responsible for the high production of IIIa' at the expense of IIIa under drier conditions. By including the 6 methyl isomer in a temperature calibration, we were able to enhance the correlation with MAT. Therefore, we conclude that 6 methyl isomers have an impact on temperature reconstructions, highlighting their inclusion in a Bale Mountain-specific temperature calibration. Using surface sediment data from Bale Mountain lakes and the East African lake database, the best performing temperature calibration is a modified MBT'$_{ME}$ including IIIa'.

With the use of the new calibration, the Garba Guracha MAT record reflects insolation variability as one of the main climatic drivers at millennial scales. Additional factors such as glacier and permafrost melting during deglaciation and the regional atmospheric circulation likely play a prominent role on shorter time scales. These additional mechanisms partly explain the asynchronicity between the Garba Guracha MAT record in the high altitude afro-alpine region of the Horn of Africa and other eastern African lake records.

Further research is necessary to understand the influences on and the origin of brGDGTs producing communities, especially at high altitudes.

**Author contribution.** LB, GGR, HFL, and MZ collected the samples. LB, CDJ, JMR and MZ developed the concept. LB and CDJ extracted, analyzed and interpreted the brGDGT data. LB led the manuscript writing with contributions and feedback from all authors. MZ acquired the funding and supervised the work.



**Competing interests**. The authors declare that they have no conflict of interest.
**Acknowledgements**
This research was funded by the German Research Council (DFG: ZE 844/10–1) in the framework of the joint Ethio-
European DFG Research Unit 2358 "The Mountain Exile Hypothesis. How humans benefited from and reshaped
African high-altitude ecosystems during Quaternary climate changes". We are grateful to the project coordination, the
Philipps University Marburg, University of Addis Abeba, the Frankfurt Zoological Society, the Ethiopian Wolf Project,
the Bale Mountains National Park, and the related staff members, especially Katinka Thielsen and Mekbib Fekadu for
their logistic assistance during our fieldwork. We thank the Ethiopian Wildlife Conservation Authority for permitting
our research in the Bale Mountains National Park.

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
