# Peer review of "A Holocene temperature (brGDGT) record from Garba"

_Biogeosciences, 2022_

## Referee Comment (RC2)

The paper by Bittner and colleagues proposes a novel mean annual air temperature (MAT) record from a high-altitude lake in Ethiopia. The record is obtained by analysis of the relative abundance of brGDGTs, bacterial membrane lipids, a proxy successfully used for more than a decade for continental paleoclimate reconstruction across the globe. In this paper, Bittner and colleagues first refines the brGDGT calibration to include specificities of northeastern African lakes before applying it to reconstruct temperature variability over the Holocene. The paper is well written and easy to follow and the research question addressed by the authors is relevant and of high interest. Indeed, the climate sensibility of the Horn of Africa has been less studied than other parts of (Eastern) Africa even though it's at the northernmost limit of the zone impacted by the ICTZ fluctuations. Overall, I would recommend the paper for publication after moderate revisions. You will find below a list of my major and minor comments on the manuscript.

Major comments:

-As a large part of the manuscript deals with the refinement of the temperature calibration, the authors should include a discussion on the origin of the brGDGT signal and what "temperature" the proxy is actually recording. Do they consider the brGDGTs to be mainly produced in the lake water column or in the catchment soils (notably in the alluvial swamp mentioned in the site description)? Can the source of brGDGTs change over time (in relation with the level of precipitations) and could this impact/bias the temperature reconstruction over the Holocene? In several instances in the discussion part, the authors seem to suggest that the lake physicochemical conditions may impact the proposed temperature record. Does that mean that they assume the brGDGTs to record the lake water temperature instead of the annual air temperature? These points need to be discussed and clarified by the authors in a revised version of the manuscript.

-I also find the discussion of the Holocene temperature record in the section 5.3 to be too superficial. I am lacking a discussion of the similarities / differences between the record presented in this manuscript and the other East African records. What do we learn from this record about the connectivity between the climate in the Horn of Africa vs. in locations closer to the equator? Or about the west/east connectivity? For example, records in Fig. 8 could be classified from the closest to the furthest from the Bale Mountains and countries should be specified for each record to help the reader orient him/herself. Also, the amplitude of temperature change is very different between the different records presented in Fig. 8 but it is not discussed by the authors.

Minor comments:

l. 85: The publication by Halamka et al. (2021; doi: 10.7185/geochemlet.213) showing production of brGDGTs triggered by oxygen limitation in an Acidobacteria should be cited and discussed here.

l. 93: Bale Mountains are cited here for the first time, without being introduced before. The authors should introduce why the Bale Mountain area is an interesting location to study the Horn of Africa temperature variability earlier in the introduction.

Material and methods: it's not clear whether the authors extracted and analysed themselves the surface sediment samples or if they just used already published data.

Figure 2, 3, S1 and S2: the colors described for the datapoints in the captions do not seem to fit the colors in the figures.

l. 208-210: I do not understand the distinction between Kenyan, Ugandan lakes and the East African ones. Are the samples named "Kenya" and "Uganda" samples from the East Africa dataset of Russell

et al. with T<10°C (higher altitudes) or are these not included in the dataset of Russell et al. and new samples analysed by the authors? This must be better explained by the authors.

l. 228-230: It is hard to draw such a hypothesis from the data distribution in the PCA. The Bale Mountain samples are vertically distributed which suggests that they have the same proportion of IIIa but it is their proportion of IIIa' (and IIb) that varies most. Moreover, it is clear in Fig. 3b and 3c that it is the East African lakes that are responsible for the good correlation between the lipid fractional abundance and the MAAT. It is difficult to see any relationship between the lipid fractional abundance and MAAT in the Bale Mountain dataset (as well as in the datasets of Kenyan and Ugandan lakes with T<10°C).

l. 235-237: and so? Is this important for the rest of the manuscript?

Figure 4: captions of A and B are inverted. The color codes in the barplot should be explained.

l. 277: no "s" at Bale Mountain.

Figure 6: again the colors mentioned in the caption do not fit with the colors of the figure.

l. 300: to be replaced by "we will only discuss"

l. 320-322: this sentence is not clear, the authors should rephrase it.

l. 347-351: are the authors suggesting that the brGDGTs are recording the lake water temperature rather than the MAT? (see my major comment above)

Section 5.3: the country where each lake is located should be added to help the reader orientate him/herself.

Fig. 8: for the stack record of Ivory and Russell, does the y-axis show absolute temperatures or delta of change?

l. 401-406: here as well, the authors suggest physical phenomena within the water column that may influence the lake water temperature. But aren't the brGDGTs supposed to be correlated with air temperature? In this regard, physical phenomena such as water stratification or ice formation should not influence the brGDGT signal (supposed to come from soil weathering within the lake catchment).

---

## Author Response (AR1)

Dear Dr. Sebastian Naeher,

we are grateful for the fast editorial process and aim to solve the minor concerns in this revision. We will address the Editor's comments concerning the three comments of Reviewer #2 point by point below.

*Dear Mr Bittner et al.,*

*Thank you for your responses to the questions and suggestions of the reviewers. I look forward to see the new version of your manuscript, which incorporates these comments and your responses.*

We incorporated all comments and responses and by doing this improved the manuscript. We carefully checked the manuscript again for clarity and edited the writing without changing the meaning. We would like to rephrase a response to Reviewer #2's comment in line 369 (now line 373) to better fit Reviewer #2's valuable comment regarding the amplitude of temperature change at Garba Guracha.

"In contrast to precipitation reconstructions based on $\delta^2$H in East Africa (Garelick et al. 2021), the temperature records do not show a clear meridional, north-south temperature change, nor an east-west pattern. The reconstructed overall temperature ranges are, however, consistent with the elevations of the lake archives. The amplitude of temperature change over the last 13 ka at Garba Guracha is ~6°C. Similar amplitudes of change have been reconstructed at other high-altitude sites (Lake Mahoma and Lake Rutundu) (Loomis et al. 2017; Garelick et al. 2022), whereas equatorial records at lower elevations yield lower temperature amplitudes (Lake Victoria and Lake Tanganyika) (Tierney et al. 2008; Berke et al. 2012), and higher temperature amplitudes are also recorded in northeast African Lake Tana (Loomis et al. 2015). In fact, Garba Guracha has some of the highest amplitude temperature changes of all of the sites during the Holocene, perhaps because it combines high elevation with a slightly higher latitude than other terrestrial African temperature records."

We also added (line 193 and 195) the equations for surface water pH from Russell et al. (2018) and the equation for conductivity published by Raberg et al. (2021).

*Additionally, I would like to ask you to please further clarify/address the following comments from Reviewer 2:*

*I don't think that you have sufficiently addressed the comments concerning Lines 228-230, so you should clarify/extend/answer this in the manuscript. You should also add the content of your response about lines 235-237 to the manuscript, because this will be useful to the readers of your manuscript. I also don't think you have replied sufficiently to the comment about Lines 401-406, so more detail would be useful to the reader.*

*Lines 228-230*

*Reviewer #2:  l. 228-230: It is hard to draw such a hypothesis from the data distribution in the PCA. The Bale Mountain samples are vertically distributed which suggests that they have the same proportion of IIIa but it is their proportion of IIIa' (and IIb) that varies most.*

Response:     The PCA illustrates changes in the complete brGDGT fingerprint. Instead, we have referred to Fig. 3A, B in the discussion, as this shows the difference in the fractional abundance of brGDGTs IIIa and IIIa' more clearly.

*Reviewer #2:*   *Moreover, it is clear in Fig. 3b and 3c that it is the East African lakes that are responsible for the good correlation between the lipid fractional abundance and the MAAT. It is difficult to see any relationship between the lipid fractional abundance and MAAT in the Bale Mountain dataset (as well as in the datasets of Kenyan and Ugandan lakes with T<10°C).*

Response:     We fully agree with Reviewer #2 that it is difficult to see the relationship between lipid fractionation and MAAT in Fig. 3. We are grateful for the feedback and apologise for overlooking this in the first round of revisions. We included an additional figure similar to Fig. 3, but focusing on lakes T>10°C, in the supplementary material and added a sentence (line 252) to the manuscript: "Narrowing the temperature range, (MAT<10 °C), the improvement remains significant: the correlation of %IIIa to MAT ($r^2$ = 0.11; p-value < 0.001) is improved by adding %IIIa' to $r^2$ = 0.31 (p-value < 0.001) (Fig. S2)." Here, we would like to add that, limiting the temperature range when reporting correlations between brGDGT proxies and temperature, will always result in low correlation coefficients, which is due to the large variability of MBT'$_{5ME}$ values encountered at sites with the same temperature. This can be seen in the global lake calibration (Martinez-Sosa et al., 2021) and global soil calibration (Dearing Crampton-Flood, 2020), and has been attributed to poorly constrained inter-site variability. As such, we argue that the significant correlation observed in the narrow temperature range here, supports the temperature dependency of the MBT'$_{5ME}$ + IIIa' ratio across a larger temperature range.

*Lines 235-237*

*Reviewer #2:*   l. 235-237: and so? Is this important for the rest of the manuscript?

Response:     We changed the sentence and moved it to line 224 "Regional differences in the brGDGT isomer abundances, especially in Bale Mountain surface sediments, are further supported by variations in the degree of cyclisation (DC') and CBT' in the eastern African lake surface sediment data (Russell et al., 2018; Baxter et al., 2019) (Fig. S2).

*Lines 401-406*

*Reviewer #2:*   *here as well, the authors suggest physical phenomena within the water column that may influence the lake water temperature. But aren't the brGDGTs supposed to be correlated with air temperature? In this regard, physical phenomena such as water stratification or ice formation should not influence the brGDGT signal (supposed to come from soil weathering within the lake catchment).*

Response:     We are sorry that we did not clarify this in the first review round. We have responded to this comment along with Reviewer #2's main comment as it addresses the question of the origin of the brGDGTs in the Garba Guracha. Evidence suggests that most of the brGDGTs in the Garba Guracha are aquatically produced and, therefore, would be influenced by physical phenomena such as water stratification and ice formation.

We add a table with the soil data to the supplements and a sentence in line 231 to further highlight this point to the reader: "The similar distribution of tetra-, penta- and

hexamethylated brGDGTs in surface sediments, illustrates a shared dominant lake-derived provenance as the East African lake dataset, as soil-derived brGDGTs are characterized by a larger fractional abundance of brGDGTs Ia (Russell et al., 2018)."

The arguments for this can be found in our first reply to the valuable comments of Reviewer #2:

"We are grateful to Reviewer #2 for highlighting this topic, which we have not addressed enough. Generally, lake water temperature and mean air temperature have a good agreement for most temperate and tropical lakes on a global scale. brGDGTs produced in the lake water column or the lake sediments reflect lake water temperature. However, due to the good agreement of MAT and lake water temperature in tropical lakes and the fact that temperatures used in the modern calibration datasets are reported in MAT and, at least in the Bale Mountains, derived from air temperature, we use MAT.

Concerning the source identification of brGDGTs, we need to highlight that the producing communities of brGDGTs are not fully understood yet. Therefore, we can only use different indicators, proxies, and comparisons to predict the origin of most brGDGTs in the Garba Guracha sediments. The Garba Guracha is/was characterised by high aquatic productivity. Several analysed proxies used for organic matter source identification point to a predominantly aquatic production ($\delta^{13}$C, TOC/N, Paq, sugar quantification ratios) (Bittner et al. 2020, 2021). Moreover, the composition of brGDGTs in soil samples and lake sediment samples in the Bale Mountain is not similar, indicating different producing communities. These findings are coherent with the results of Russell et al. (2018) that brGDGTs in lake sediments of eastern Africa are dominantly lake-derived. Therefore, we suggest that also most brGDGTs in the Garba Guracha sediment archive are of aquatic origin.

To account for the valuable feedback of Reviewer #2 we added a figure and a table to the supplements and a paragraph (line 262) to the manuscript focussing on the origin of brGDGTs: "In general, the sediments of the Garba Guracha are characterised by a high input of aquatic organic matter. Several analysed proxies used to identify the source of organic matter indicate a predominantly aquatic production ($\delta^{13}$C, TOC/N, $P_{aq}$, sugar quantification ratios) (Bittner et al., 2020, 2021). The composition of brGDGTs in the sediment of Lake Garba Guracha is inconsistent with the soil samples in Bale Mountain, indicating different producing communities (Fig. S3, Table S5). These findings are concurrent with the results of Russell et al. (2018) that brGDGTs in eastern African lake sediments are dominantly lake-derived. Therefore, we suggest that most brGDGTs in the Garba Guracha sediment archive are of aquatic origin.""

*Please also address the comment from the Copernicus Editorial Team about your map (Figure 1).*

*With the great help of the Copernicus Editorial Team we added* to Figure 1: "© OpenStreetMap contributors 2019. Distributed under the Open Data Commons Open Database License (ODbL) v1.0." and "© Microsoft " for the Bing Map Image.

We also sent this change to the Copernicus editorial team on 04.05.2022. As we were not asked to change anything else, we felt that this solved the comment.